# Exploratory study evaluating the relationships between perinatal adversity, oxidative stress, and infant neurodevelopment across the first year of life

**Kameelah Gateau**[1,2]*, **Lisa Schlueter**[3], **Lara J. Pierce**[4], **Barbara Thompson**[5], **Alma Gharib**[2,3], **Ramon A. Durazo-Arvizu**[1,2], **Charles A. Nelson**[6,7,8], **Pat Levitt**[1,2,4]

**1** Department of Pediatrics, Keck School of Medicine of University of Southern California, Los Angeles, California, United States of America, **2** Children's Hospital Los Angeles, Los Angeles, California, United States of America, **3** Developmental Neuroscience and Neurogenetics Program, The Saban Research Institute, Los Angeles, California, United States of America, **4** York University, Department of Psychology, Toronto, ON, Canada, **5** Department of Pediatrics and Human Development, Michigan State University, Grand Rapids, Michigan, United States of America, **6** Department of Pediatrics, Division of Developmental Medicine, Boston Children's Hospital, Boston, Massachusetts, United States of America, **7** Harvard Medical School, Boston, Massachusetts, United States of America, **8** Harvard Graduate School of Education, Boston, Massachusetts, United States of America

* kgateau@chla.usc.edu

**Data Availability Statement:** All relevant data are publicly available from the Dryad repository https://

## Abstract

Early childhood adversity increases risk for negative lifelong impacts on health and wellbeing. Identifying the risk factors and the associated biological adaptations early in life is critical to develop scalable early screening tools and interventions. Currently, there are limited, reliable early childhood adversity measures that can be deployed prospectively, at scale, to assess risk in pediatric settings. The goal of this two-site longitudinal study was to determine if the gold standard measure of oxidative stress, F2-Isoprostanes, is potentially a reliable measure of a physiological response to adversity of the infant and mother. The study evaluated the independent relationships between F2-Isoprostanes, perinatal adversity and infant neurocognitive development. The study included mother-infant dyads born >36 weeks' gestation. Maternal demographic information and mental health assessments were utilized to generate a perinatal cumulative risk score. Infants' development was assessed at 6 and 12 months and both mothers and infants were assayed for $F_2$-isoprostane levels in blood and urine, respectively. Statistical analysis revealed that cumulative risk scores correlated with higher maternal ($p = 0.01$) and infant ($p = 0.05$) $F_2$-isoprostane levels at 6 months. Infant $F_2$-isoprostane measures at 2 months were negatively associated with Mullen Scales of Early Learning Composite scores at 12 months ($p = 0.04$). Lastly, higher cumulative risk scores predicted higher average maternal $F_2$-isoprostane levels across the 1-year study time period ($p = 0.04$). The relationship between perinatal cumulative risk scores and higher maternal and infant $F_2$-isoprostanes at 6 months may reflect an oxidative stress status that informs a sensitive period in which a biomarker can be utilized prospectively to reveal the physiological impact of early adversity.

datadryad.org/stash/share/NGyy8lrN91J3fiokLwl
YFKyBc3OMWY6TYw1BS4bQuiI.

**Funding:** This study was financially supported by
Simms/Mann Institute & Foundation (https://www.
simmsmanninstitute.org) in the form of salary, and
in the form of an award (Simms/Mann Next
Generation Research Fund) received by PL. There
are no grant or award numbers associated with
this funding to declare. This study was also
financially supported by Keck School of Medicine
of USC (https://keck.usc.edu/) in the form of salary
for PL. The specific roles of this author are
articulated in the 'author contributions' section.
This study was also financially supported by The
Center on the Developing Child (Harvard
University) in the form of an award received by PL
and CN. There are no grant or award numbers
associated with this funding to declare. The
funders had no role in study design, data collection
and analysis, decision to publish, or preparation of
the manuscript.

**Competing interests:** The authors have read the
journal's policy and have the following competing
interests: PL is an employee of Simms/Mann
Institute & Foundation (https://www.
simmsmanninstitute.org). PL is also an employee
of Keck School of Medicine of USC (https://keck.
usc.edu/). This does not alter our adherence to
PLOS policies on sharing data and materials. There
are no patents, products in development or
marketed products associated with this research to
declare.

## Introduction

At a population level, experiencing adversity early in childhood has been associated with long term mental and physical health morbidities [1, 2]. The Adverse Childhood Experiences (ACES) study established a link between seven domains of adversity (psychological, physical or sexual abuse, parental violence, exposure to substance abuse, mentally ill or imprisoned household members) and adult health risk behaviors and chronic illness [1]. Since that study, there have been numerous reports that have validated the associations that are described at a population level. Nationally,15–20% of all individuals report having experienced 4 or more early adversity events that are associated with a 20-year reduction in total lifespan [2–4]. Chronic adult diseases that are associated with ACES generate the most healthcare costs [5]. Additionally, among the leading causes of death (cancer, heart disease and suicide), 15% of mortality is attributable to experiencing childhood adversity (CA) [2]. While the link between adversity and poorer health outcomes has been established, recent studies report that current ACES screening at the individual level are not predictive of later life disease risk [6, 7]. Additionally, the psychological challenges, including stress, anxiety and depression, and socioeconomic instability due to the COVID 19 pandemic has been reported to increase the risk of a child experiencing early life adversity and its downstream effects [8–10]. Though the COVID-19 pandemic is both an acute and pervasive potential source of CA, recent findings continue to rely on retrospective reporting. Thus, despite the heightened concern for CA in this current public health crisis, there are not reliable and prospective scalable measures of early adversity.

Picard et al established the prevailing hypothesis linking early CA to disruption of typical development, theorizing that chronic stress disrupts homeostasis that generates "allostatic load" on the brain and body physiological systems [11, 12]. Not all stressors are equivalent, and when they are brief or isolated and buffered by support systems the physiologic response is acute (positive or tolerable stress responses) and recovery to physiological homeostasis occurs [11, 13, 14]. However, a toxic stress response occurs when a child faces multiple, concomitant and severe stressors with limited socioemotional buffers [13]. For example, when some children experience multiple adverse environmental and psychosocial risk factors, there appears to be a higher likelihood of psychological harm and psychiatric disorders[15, 16]. Developmental literature too has demonstrated that multiple adverse risk factors experienced in conjunction with one another lead to poorer cognitive outcomes and educational achievement [17]. For example, the findings from the Bucharest Early Intervention Project highlight how the absence of crucial early social engagement and environmental inputs for institutionalized children were associated with impaired executive functioning and cognitive abilities, language deficits and higher prevalence of psychological disorders [18–21].

Physiologically, the toxic stress response consists of a derangement of the normal neuro-endocrine-immune interactions, alterations of mitochondrial function and oxidative stress, with failure of the body to normalize these changes after the stressor is removed [22–25]. Oxidative stress is hypothesized to be one of the major homeostatic dysregulations contributing to allostatic load and toxic stress response [25]. Oxidative stress occurs when there is an imbalance of reactive oxidative species (ROS) and the neutralizing capacity of antioxidant systems, leading to disruption of normal cellular function [12, 24]. The brain is particularly susceptible to ROS [26].

Both animal and human studies have demonstrated the association between exposure to early life adversity and oxidative stress [27–31]. In mice, it has been demonstrated that early life stress alters mitochondria and neurodevelopmental proteins, leading to disruption of mitochondrial energetics [24]. The relationship between oxidative stress and neurodevelopmental outcomes was hypothesized by Ross and colleagues. Ross postulated that oxidative stress and elevated ROS was the link between low levels of poly-unsaturated fatty acids and elevated

phospholipase A2 levels (the enzyme that cleaves damaged fatty acids from the cell membrane) seen in patients with neurodevelopmental disorders [32]. In a recent study elevated maternal oxidative stress in the third trimester of pregnancy was associated with social impairments in their children at 4 years of age [31]. Another recent study demonstrated an association of CA, oxidative stress, and mental health in adolescent girls in the juvenile justice system [33]. This study found a significant association between elevated oxidative stress marker $F_2$-isoprostane, internalizing disorders (generalized anxiety disorder, depressive disorder, bulimia and anorexia nervosa, dysthymic disorder) and attention deficit hyperactivity disorder.

Studies have demonstrated that $F_2$-isoprostane (8-IsoF2a) serves as a sensitive, gold standard biomarker of oxidative stress [34, 35]. 8-IsoF2a metabolites are generated as the result of peroxidation of arachidonic acid by ROS and can be quantified in urine and blood serum samples [34, 36, 37]. At the time of birth, newborns exhibit a physiologic oxidative stress response as they transition from a low oxygen environment *in utero* to a relatively high oxygen environment. Freil and colleagues demonstrated that serum 8-IsoF2a levels in infants at birth are markedly elevated and decline precipitously over the first 6 months of life to stable, adult levels [38]. Importantly, the elevated 8-IsoF2a levels in infants at birth are not influenced by 8-IsoF2a levels during pregnancy, with no indication of transplacental transference [39]. With established normative values, being easily assessable, and no vertical transmissibility, 8-IsoF2a is an ideal potential candidate biomarker of early adversity.

Dimensional approaches to capture early adversity provide opportunities to both identify elements of experiences that are most potent in derailing homeostasis and characterize the interdependent nature of these factors. Approaching quantification and assessment of early adversity from a dimensional wholistic approach should also include consideration of the screening window. Studies both in animals and humans have demonstrated the association of maternal stress during pregnancy and altered neurocognitive development and stress responses in offspring, highlighting a crucial screening window existing in the perinatal (pregnancy and first postpartum year) time periods[40–43]. A review done by Van den Bergh et al. reported on seventeen studies that evaluated the relationship between maternal perinatal stress and neurodevelopmental outcomes [41]. This review found that perinatal stress was associated with cognitive, behavioral and emotional problems in infants, preschool and adolescent children. Work done by Silvera and colleagues evaluated cumulative perinatal adversity's association with developmental outcomes. This study found that higher perinatal adversity scores were significantly associated with lower neurodevelopmental and school readiness scores and increased behavioral problems [43]. These important findings not only infer a crucial perinatal screening period they also highlight the long-lasting effects experiencing early adversity has on neurodevelopmental trajectories.

When forecasting scalability of early adversity screening, an understanding of the mechanisms underlying the impact of early adversity on neurodevelopmental outcomes is also needed. Our mediation model, adapted from Horn et al, presented in Fig 1, hypothesizes a mechanism of how perinatal adversity may impact infant neurodevelopment [33]. To our knowledge no studies to date have prospectively evaluated the relationships between perinatal adversity, oxidative stress, and neurocognitive development in infants. Given the limited sample size we provide exploratory data analyses to evaluate the viability of the mediation model proposed in Fig 1. In this study, we aimed to evaluate the individual associations between perinatal adversity, oxidative stress as measured by 8-IsoF2a and infant cognitive development.

## Methods

### Ethics statement

Ethical approval for the study was obtained from Institutional Review Boards (IRB) at both sites (Children's Hospital Los Angeles Institutional Review Board and Boston Children's

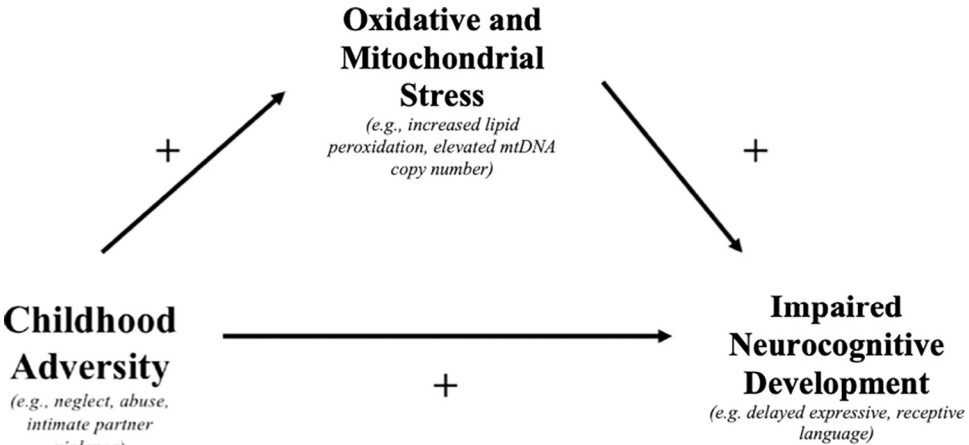

**Fig 1. Adapted mediation model of early childhood adversity, oxidative stress and neurocognitive development [33].**

Hospital Institutional Review Board). Informed written consent was obtained from the mother for all mother- infant dyads who participated in the study.

## Participants

The study included 116 mother- infant dyads recruited between 2016–2019 from pediatric practices affiliated with Boston Children's Hospital and Children's Hospital Los Angeles. Participants were recruited if they met the following inclusion criteria: receiving early postnatal services at the designated primary care clinics, mothers aged 18 or older with an infant $< 2$ months, and birth weight greater than the 20th percentile (at least 2,500 grams and typically developing).

Exclusion criteria included infants who were born prematurely ($< 36$ weeks of gestation); infants who had congenital disorders (genetic, metabolic, malformations, or neurological disorder etc.); infants who experienced birth complications (fetal distress, requiring resuscitation at birth, meconium aspiration syndrome, *etc*). Study visits took place at 2, 6, 9, and 12 months from 2016 to 2019. All deidentified study data were entered into REDCap by the research coordinator. No author had access to information that could identify individual participants during or after data collection.

## Measures

**Family demographic factors.** The family demographic factors were collected through self-report from mothers at the 2- month visit. Demographic factors of interest were marital status, highest level of education completed, range of annual family income, age, and racial identity (which we recognize as a social construct). Marital status was represented as single (single, widowed, separated, divorced) or partnered (married, cohabiting). Maternal education was assessed as seven categories ranging from 8th grade or less to M.D., Ph.D., J.D., or equivalent. The nine categories of annual family income ranged from $<\$5,000$ to $\$100,000+$. Maternal age was self-reported at the time of the 2-month visit. Participants were asked to select a racial category from a selection of 14 different identities which included the following options: White, Black or African American, American Indian or Alaska Native, Asian Indian, Chinese, Filipino, Japanese, Korean, Vietnamese, Other Asian, Native Hawaiian, Guamanian or Chomorro, Samoan, and Other Pacific Islander. Racial information collected about the study participants in Boston only included the infants, thus the race information analyzed in this study are of the race designations of the infants as reported by their mothers.

**Maternal mental health.** Two assessments of maternal mental health were utilized in this study. *The Edinburgh Postnatal Depression Scale* (EPDS) is a validated 10-item screening tool frequently used in the perinatal and postnatal setting to identify mothers with depression [44]. Depressive symptoms are assessed using a four-tiered severity scale for the prior 7 days from the research visit. Scores range from 0 to 30, with a score of 13 or greater indicating risk for major depression [44, 45]. The EPDS was collected at 2 months postpartum. *The Patient Health Questionnaire 9* (PHQ9) is a 9-item assessment used to identify and monitor over time major depressive symptoms in the general population and in the postpartum period [46, 47]. This assessment utilizes a four-tiered scale assessing the frequency of depressive symptoms over the past 2 weeks. The score ranges from 0 to 27, with a score of 10 or greater indicating elevated depressive symptoms. The PHQ9 was completed at 2 months postpartum. Both measures were used in this study to assess both anxiety and depressive symptoms (EPDS) and the constitutional symptoms of depression (PHQ9) [47].

## Cumulative risk score

The cumulative risk score (CRS) assessing perinatal adversity was calculated using the cumulative risk modeling described by Evans et al [48]. Demographic factors including age, highest education attained, marital status, family income and race were standardized into risk categories and calculated as being present (1) or absent (0). Risk delineations for each demographic factor were calculated based on a participant occupying the lowest standard deviation of the sample mean (age< = 25 years, income <$25,000) or occupying a demographic category previously described as having higher risk for infant morbidity and mortality (Black race, < = High School education, single marital status) [49–51]. Approximately 30% of participants did not provide income data. Rates of missing data did not differ across study sites. For those participants with missing income data, risk stratifications were assigned using neighborhood poverty levels and the Internal Revenue Services' census tract definition of low-income communities (LIC) [52–54]. Mental health risk scaled scores were generated from previously described risk stratified scoring guidelines for PHQ9 and EPDS. Scores were standardized as low/no risk (0), medium risk (1) or high risk (2) [44–47]. Total cumulative risk score (CRS) for each participant was calculated by summation of all demographic and mental health risk categories occupied to create an overall perinatal cumulative risk score, with a range of cumulative risk scores from 0–9.

## Cognitive scores

Infant cognition was measured using The Mullen Scales of Early Learning (MSEL). The MSEL is a play-based developmental assessment that evaluates gross motor, visual reception, fine motor, expressive language, and receptive language [55]. An early learning composite score (ELC) is generated from the assessments excluding gross motor (excluded per standard practice). This composite score is then standardized to compare cognitive developmental status from birth through sixty-eight months of age and has a mean of 100 and standard deviation of 15. [55]. Assessments were done at 6 and 12 at both study sites by trained and assessment-validated members of the research team in either English or Spanish, based on the primary language spoken at home.

## Oxidative markers

The lipid oxidation biomarker $F_2$-isoprostane (8-IsoF2a) was the oxidative marker of interest. 8-IsoF2a is one of several isoprostane products generated by reactive oxygen species peroxidation of arachidonic acid [32, 33]. Infant urine samples were collected via bag specimen at all 4

study visits, aliquoted and stored frozen at –80˚C within 1 hour of collection. For urine, adjustments were made for potential concentration differences by measuring creatinine in each sample. No other exogenous or endogenous factors have been consistently shown to effect urinary 8-IsoF2a levels thus no other adjustments were made [56]. Concentrations were expressed as ng/mg creatinine. Maternal blood samples were collected by a trained phlebotomist at all 4 study visits. Urine and blood samples were shipped in batches, deidentified with a subject code, to the Eicosanoid Core Laboratory at Vanderbilt University for 8-IsoF2a analyses. 8-IsoF2a was measured using gas chromatography/negative ion chemical ionization mass spectrometry, following previously published methods [34, 36, 57]

## Statistical analysis

Demographic characteristics of both mothers and infants, infant MSEL scores and maternal and infant 8-IsoF2α levels were compared by site using chi-squared analysis for discrete categorical variables and independent student t-test for continuous variables. Due to skewing in the data, EPDS and PHQ9 scores were compared using a Mann-Whitney U test. CRS between both sites were evaluated by independent student t-test. *A priori* α to distinguish differences between sites was set as 0.05.

A sub-analysis by study location was done to evaluate the participants with missing income data. Chi-squared analysis was utilized to evaluate site differences in missing income data. Demographic, EPDS and PHQ9 variables for participants with missing income data were evaluated in comparison to individuals with income data as described above. Pearson Correlation Coefficients were calculated to evaluate the association between LIC census tract data and participants with reported income.

**Perinatal adversity and oxidative stress** Pearson Correlation Coefficients were used to evaluate the bivariate associations between infant and maternal 8-IsoF2α and CRS. The effect of perinatal adversity on oxidative stress was also assessed by fitting a linear mixed-effects model (LME) with 8-IsoF2α over the first year of life (measured at 2, 6, 9, and 12 months) as the dependent variable and CRS as the independent variable. LME models take advantage of the within-individual correlation and make it possible to include research participants with incomplete observations under robust statistical assumptions.

**Oxidative stress and infant neurocognitive development** The bivariate associations between infant and maternal 8-IsoF2α and MSEL scores at 6 and 12 months were assessed via Pearson Correlation Coefficients and Linear regression model that measured the change of MSEL scores between 6 and 12 months, and the change of 8-IsoF2α during the same period.

**Perinatal adversity and infant neurocognitive development** The bivariate associations between MSEL and CRS were assessed via Pearson Correlation Coefficients. The change in the MSEL score from month 6 to month 12 was regressed on the CRS score to test the association between perinatal adversity and infant cognitive development. All analyses were performed using SPSS Version 27.0 (IBM Corp, Armonk, NY, USA) and STATA version 17.1 (College Station, TX).

## Results

There were 116 mother-infant dyads recruited across both study sites. Six mother-infant dyads who did not meet inclusion criteria following enrollment were excluded. Developmental outcomes on infants with MSEL scores at 6 (n = 96) or 12 months (n = 86) were included in the analysis. Demographic characteristics and developmental scores for infants are presented in Table 1 and demographic characteristics and mental health scores seen in the mothers are presented in Table 2. Notably, there was a statistically significant difference (P<0.001) in race across the two

**Table 1. Infant demographics and developmental scores.**

| | Los Angeles | | Boston | | p |
| | n = 55 | | n = 55 | | |
| | N | % | N | % | |
|---|---|---|---|---|---|
| **Gestational Age** | | | | | |
| <37 post menstrual age | 6 | 11% | 2 | 4% | 0.16 |
| >37 post menstrual age | 49 | 89% | 50 | 91% | |
| **Sex** | | | | | |
| Male | 32 | 58% | 28 | 51% | 0.57 |
| Female | 23 | 42% | 27 | 49% | |
| **Birth Weight (grams)** | 3359* | 522^ | 3290* | 420^ | 0.46 |
| **Birth Length (cm)** | 51* | 3^ | 49* | 8^ | 0.35 |
| **Oxygen Required** | 1 | 1% | 1 | 1% | |
| **NICU Stay** | | | | | |
| Yes | 3 | 6% | 0 | 0% | *** |
| # of days | 4.3* | 3.1^ | 0 | 0 | *** |
| **Race** | | | | | |
| White | 43 | 78% | 11 | 20% | <0.001 |
| Black or African American | 3 | 6% | 31 | 56% | |
| Chinese | 0 | 0% | 1 | 2% | |
| Filipino | 1 | 2% | 0 | 0% | |
| Vietnamese | 0 | 0% | 1 | 2% | |
| Biracial or Multiracial | 3 | 6% | 8 | 15% | |
| I do not wish to disclose | 5 | 9% | 3 | 6% | |
| **Mullen Scales of Early Learning ELC** | | | | | |
| 6 month (n = 93) | 86 | 82–89^ | 94 | 91–97^ | 0.001 |
| 12 month (n = 86) | 87 | 85–91^ | 102 | 100–106^ | <0.001 |

\* mean

^ 95% confidence interval

\*\*\* not asked or collected

study sites with 78% of infants identified as White (which includes Hispanic and Non-Hispanic ethnicities) in Los Angeles and 56% identified as Black or African American in Boston (Table 1). As previously published (Valdes et al. 2020), there were site differences in MSEL scores at both 6 and 12 months with higher scores for the Boston population for both time points. Additionally, maternal demographics differed between the 2 study sites, including family income (p = 0.01), marital status (p = 0.03), and primary language spoken at home (p<0.001) (Table 2) [52]. The median scores for mental health assessments were 4 (IQR 1–7) for EPDS and 2 (IQR 0–4) for PHQ9, with no differences in scores between the two sites for either measure (Table 2).

A maximum CRS of 8 was seen (out a maximum achievable score of 9) and there were no significant differences in mean scores between sites; with Boston having a mean CRS of 2.8 (SD 1.9) and Los Angeles having a mean CRS of 2.7 (SD 1.3) (p = 0.65). Analysis of reported income and LIC census tract data demonstrated a significant correlation between living in a LIC and a reported income of less than $25,000 (r = 0.36, p<0.001). Sub-analysis of the 32 participants with missing income data demonstrated 14% were from Boston and 15% were from Los Angeles. For participants with missing income data, there were no statistically significant differences in demographic, EPDS or PHQ 9 variables, except infant race (p = 0.001), which is consistent with results of the aggregate analysis.

**Table 2.** Maternal demographics and mental health scores.

| | Los Angeles n = 55 | | Boston n = 55 | | p |
|---|---|---|---|---|---|
| | N | % | N | % | |
| **Maternal Education** | | | | | |
| 8th grade or less | 4 | 7% | 1 | 2% | 0.06 |
| Some high school | 11 | 20% | 6 | 11% | |
| High school/GED | 29 | 53% | 24 | 44% | |
| Associate degree | 4 | 7% | 12 | 22% | |
| Bachelor's degree | 4 | 7% | 4 | 7% | |
| Master's degree | 3 | 6% | 4 | 7% | |
| MD, JD, PhD or equivalent | 0 | 0% | 1 | 2% | |
| I do not wish to disclose | 0 | 0% | 3 | 6% | |
| **Family Income** | | | | | |
| Less than 5,000 | 9 | 16% | 5 | 9% | 0.01 |
| 5,000 through 11,999 | 3 | 6% | 4 | 7% | |
| 12,000 through 15,999 | 9 | 16% | 1 | 2% | |
| 16,000 through 24,999 | 5 | 9% | 3 | 6% | |
| 25,000 through 34,999 | 5 | 9% | 9 | 16% | |
| 35,000 through 49,999 | 5 | 9% | 5 | 9% | |
| 50,000 through 74,999 | 2 | 4% | 3 | 6% | |
| 75,000 through 99,999 | 0 | 0% | 6 | 11% | |
| 100,000 or greater | 0 | 0% | 4 | 7% | |
| I don't know | 10 | 18% | 11 | 20% | |
| I do not wish to disclose | 7 | 13% | 4 | 7% | |
| **Marital Status** | | | | | |
| Single | 23 | 42% | 34 | 62% | 0.03 |
| Married | 24 | 44% | 11 | 20% | |
| Cohabiting | 6 | 11% | 5 | 9% | |
| Did not respond | 2 | 4% | 5 | 9% | |
| **Maternal Age** | | | | | |
| 18–25 | 18 | 33 | 17 | 31% | 0.63 |
| 26–30 | 18 | 33% | 18 | 33% | |
| 31–35 | 8 | 15% | 14 | 26% | |
| 36–40 | 8 | 15% | 6 | 11% | |
| 40–45 | 3 | 6% | 0 | 0% | |
| **Maternal Race** | | | | | |
| White | 42 | 76% | *** | *** | *** |
| Black or African American | 2 | 4% | *** | *** | |
| Filipino | 1 | 2% | *** | *** | |
| Biracial or multiracial | 3 | 6% | *** | *** | |
| I do not wish to disclose | 7 | 13% | *** | *** | |
| **Primary language at home** | | | | | |
| English | 14 | 26% | 44 | 80% | <0.001 |
| Spanish | 27 | 49% | 4 | 7% | |
| Both | 13 | 24% | 0 | 0% | |
| Other | 1 | 2% | 6 | 11% | |
| **Edinburgh Postnatal Depression Scale** | | | | | |
| | 3.0** | 0–6^^ | 4.0** | 2–7^^ | 0.09 |

(*Continued*)

**Table 2.** (Continued)

| | Los Angeles | | Boston | | p |
|---|---|---|---|---|---|
| | n = 55 | | n = 55 | | |
| | **N** | **%** | **N** | **%** | |
| **Patient Health Questionnaire 9** | | | | | |
| | 1.0** | 0–4^^ | 2.0** | 1–4^^ | 0.42 |

** median

^^ Interquartile Range

The bivariate correlation coefficients depicted in Table 3 represent the individual associations of maternal and infant 8-IsoF2α, cumulative risk score and infant MSEL scores at 6 and 12 months. Elevated 8-IsoF2a levels were significantly correlated with elevated cumulative risk scores at 6 months in both infants (p = 0.05) and mothers (p = 0.01). Additionally, higher infant 8-IsoF2a levels at 2 months were significantly associated with lower MSEL scores at 12 months (p = 0.04). Cumulative risk scores were not statistically significantly correlated with infant MSEL scores at 6 or 12 months.

To date, there is a lack of data on changes of 8-IsoF2a levels over the first postnatal year for infants and their mothers serving as primary caregivers. The longitudinal data collected in the present study provided an opportunity to perform analyses using a mixed effects model strategy. First, Fig 2 reports significant differences of 8-IsoF2a levels over time for both infants and mothers (p<0.001). At 2 months, higher infants 8 -IsoF2α levels were observed for infants located in Los Angeles (7.1 ng/mg Cr) compared to Boston (4.1 ng/mg Cr; p< 0.01) (Fig 2A). There were no differences in infant 8-IsoF2a levels between sites at the other time points (6, 9 or 12 month visit, p > 0.05 for each comparison). At 9 months, higher maternal 8-IsoF2α levels were observed for mothers located in Los Angeles (0.03 ng/dl) compared to Boston (0.02 ng/dl; p<0.001) (Fig 2B). No differences for maternal 8-IsoF2a levels were observed between sites at the other time points (2, 6, or 12 month visit).

Second, the linear mixed effects model (MLE) evaluated the association of CRS with maternal and infant 8-IsoF2a (Table 4). Notably, higher CRS predicted higher average maternal

**Table 3. Pearson correlations of maternal and infant 8-IsoF2a, cumulative risk scores, and infant MSEL scores.**

| | Maternal 8-IsoF2a | | | | | |
|---|---|---|---|---|---|---|
| **Study Visit** | **Perinatal Cumulative Risk Score** | | **MSEL 6 months** | | **MSEL 12 months** | |
| | **r** | **p** | **r** | **p** | **r** | **p** |
| 2 months | 0.14 | 0.24 | 0.13 | 0.32 | -0.17 | 0.2 |
| 6 months | 0.31 | 0.01 | -0.07 | 0.59 | -0.12 | 0.4 |
| 9 months | 0.04 | 0.75 | ***** | ***** | -0.26 | 0.1 |
| 12 months | -0.01 | 0.93 | ***** | ***** | -0.10 | 0.5 |
| | Infant 8-IsoF2a | | | | | |
| | r | p | r | p | r | p |
| 2 months | -0.09 | 0.45 | -0.17 | 0.2 | -0.27 | 0.04 |
| 6 months | 0.25 | 0.05 | 0.14 | 0.3 | 0.35 | 0.01 |
| 9 months | -0.14 | 0.39 | ***** | ***** | -0.06 | 0.7 |
| 12 months | -0.17 | 0.34 | ***** | ***** | -0.01 | 0.96 |

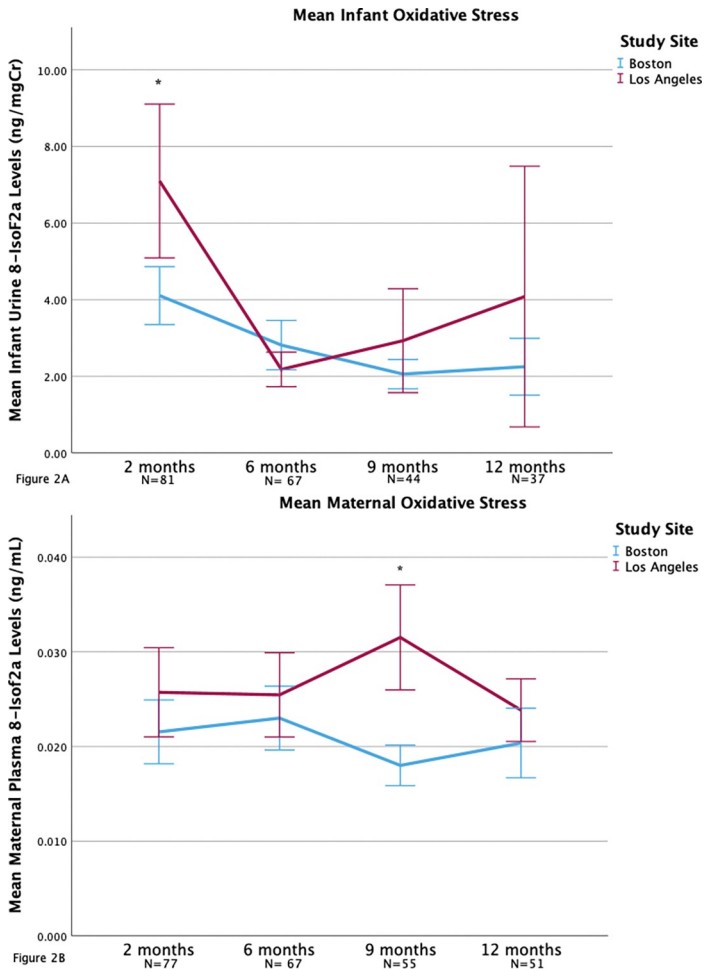

**Fig 2. Mean 8-IsoF2a levels across the first year of life.** (A) Infant 8-IsoF2a. (B) Maternal 8-IsoF2a. Error bars represent 95% confidence interval. *p<0.01.

8-IsoF2a across the 1-year study time period. When adjusting the model for each individual time point higher CRS still predicted significantly higher maternal 8-IsoF2a regardless of the study time point. CRS, however, did not predict infant 8-IsoF2a levels (Table 4). Regression models evaluating the relationship between maternal and infant 8-IsoF2a & infant MSEL and CRS & MSEL were not statistically significant.

## Discussion

The primary goal of the present exploratory study was to evaluate the potential individual associations between perinatal adversity, maternal and infant oxidative stress and infant cognitive

**Table 4. Linear mixed effects models of CRS as a predictor of maternal and infant 8-IsoF2a across the first year of life.**

|  | Maternal 8-IsoF2a | | | | Infant 8-IsoF2a | | | |
|---|---|---|---|---|---|---|---|---|
|  | **B** | **SE** | **z** | **p** | **B** | **SE** | **z** | **P** |
| **Intercept** | 0.02 | 0.002 | 12.28 | <0.001 | 4.01 | 0.54 | 7.56 | <0.001 |
| **CRS** | 0.001 | 0.001 | 2.07 | 0.04 | -0.17 | 0.22 | -0.76 | 0.45 |
| **CRS*Time** | 0.001 | 0.001 | 2.04 | 0.04 | -0.29 | 0.22 | -0.79 | 0.43 |

development as measured by MSEL composite scores. For this study, we generated a cumulative-risk score (CRS), using a previously published strategy to capture perinatal adversity through assessment of maternal mental health scores and demographic factors [48]. Statistical analyses revealed that higher CRS were associated with higher infant and maternal 8-IsoF2a levels at 6 months. Additionally, higher infant 8-IsoF2a levels at 2 months were associated with lower MSEL scores at 12 months. Lastly, higher CRS significantly predicted higher average maternal 8-IsoF2a levels over the first year of life. No association was observed between CRS and infant cognitive scores. We note that there are limitations in interpreting CRS, due to the lack of inclusion of structural and socio-economic factors, as well as measures of resilience.

Despite these challenges, the current study introduced for the first time a measure that is more proximate to hypothesized adaptive physiological processes that may occur due to the experience of early adversity. Oxidative stress, mediated through ROS has been hypothesized to be involved in perturbations of the stress response that may subsequently yield a toxic stress response [22, 23, 25, 35–37]. There have been numerous studies that have linked elevated levels of 8-IsoF2a to adult morbidities including heart disease and depression [58, 59]. Pediatric studies have found an association between elevated 8-IsoF2a with growth restriction and poor respiratory and neurodevelopmental outcomes [31, 33, 60, 61]. There have been no prospective longitudinal studies in the perinatal and postpartum periods, however, that have evaluated the association between early experiences of adversity, oxidative stress and infant neurodevelopment. Measures of potential oxidative stress in primary caregivers also is a novel dimension of the current study.

The findings in the present study showing that infant 8-IsoF2a levels decrease by 6 months and then subsequently increase, together with the correlation of elevated cumulative risk and elevated 6-month infant 8-IsoF2a levels are different from previously established temporal changes of oxidative stress. Infants at birth have a physiologically elevated ROS burden that is 4-fold higher than normal adult values, due to transitioning from a relatively hypoxic environment to a hyperoxic environment [38]. The levels of ROS during the first 6 months normally drop precipitously to adult levels [38]. In this study, however, levels between 2 and 6 months decrease but then increase. Additionally, the finding that higher CRS predicted higher maternal 8-IsoF2a across first year postnatal may indicate the utility of 8-IsoF2a as a longitudinal marker of a toxic stress response. The data captured in the present exploratory study suggest that future studies should include biomarkers of metabolic disruption early in the postnatal period to validate the sensitivity and specificity of this measure.

Interestingly, while higher 2 month infant 8-IsoF2a were significantly correlated with lower MSEL scores at 12 months, higher 6 month infant 8-IsoF2a were significantly correlated with higher MSEL scores at 12 months. This lone finding may due be in part to the 6-month time point being the physiologic nadir of oxidative stress and given most infants had lower levels of 8-IsoF2a this finding may be driven by a few individuals. Ongoing studies by this group with larger more diverse sample sizes are likely to provide more information about these correlations. Additionally, there was not a statistically significant association between the CRS and MSEL scores at 6 and 12 months. This may be due in part to the limited number of high cumulative risk dyads in this study population. The mean CRS was approximately 2.7 out of a possible 9, indicating that a broader distribution in future studies is warranted. Such a study is underway, including a larger, more diverse sample. It is noteworthy that previous studies reporting associations between cumulative risk and developmental outcomes performed their earliest developmental evaluation at 24 months [43]. Further analysis of the developmental assessments and 8-IsoF2a levels of current dyads at later ages may reveal more robust relations with demographic and maternal mental health risk scores.

The exploration of a physiologic biomarker of a toxic stress response lends to the possibility of identifying a more informative measure of the homeostatic disruption that adversity confers. The self-report survey methods that anchor current adversity screening pose many challenges with forecasting individual health states, particularly in the pediatric period [6]. Many of the most widely used screening tools assess the most severe forms of adversity, limiting the detection of more subtle forms or variable responses to adversity [62]. Additionally the current screening surveys were developed from retrospective adult studies that focus on familial and household dysfunction with little to no assessment of other forms of stress and adversity (i.e. bullying, discrimination, economic adversity etc.) [7]. In the present study, adversity was assessed through a cumulative risk score that included economic, psychosocial, and severity graded mental health factors. These factors are important in understanding the structural and subjective context in which the mother-infant dyad exist, however, these assessments do not provide insight into the individual's physiological response to these contexts. The multi-finality within the risk and adversity literature is often discussed in terms of buffering and resilience. More specifically, a toxic stress response occurs when a child faces multiple, concomitant, and severe stressors with limited socioemotional buffers available. Self-reported screening surveys, especially those that focus solely on adversity items and fail to include measures of buffering and resilience, alone are limited in their ability to reliably predict later individual negative outcomes. In contrast, biomarkers offer critical insight into the individual's physiological responses to a myriad of risk and resilience factors. Additionally biomarkers have been shown to better predict health outcomes when compared to self-reported health status [63]. Thus, we suggest that the goal to reliably capture early risk due to childhood adversity should include both sensitive and specific objective biological measures.

There are many limitations to this study. First, we interpret the results with caution due to the limited sample size and relatively low prevalence of reported or endorsed cumulative risk in the study population. Second, the developmental window of 2–12 months is early, and does not provide evidence for stability of these findings. Thus, the impact of perinatal adversity on developmental trajectories at later ages is needed. This will be addressed in future studies as data from later ages are collected and analyzed. Third, income data for thirty percent of participants were not available and the risk designations for that variable were gathered from neighborhood poverty data, which cautions interpretability of the measured associations. Additionally, the cumulative risk score income and age cut off values were based on the observed distribution within the study population, which limits the generalizability of the score. Fifth, the maternal-infant relationship was also not assessed or included in the analysis constraining the ability to directly assess a toxic stress response in this sample. There was incomplete data on self-reported maternal race. Infant race therefore was used but may result in incomplete data capture when maternal and infant race is not concordant. Lastly, race is a social construct and a poor surrogate for capturing the true risk factors that inform the negative health outcomes that are experienced by historically marginalized individuals [64]. We recognize that more nuanced and wholistic measures are needed to capture the environment and experiences of individuals assigned or self-identifying in marginalized racial categories.

With these limitations in mind, the study did identify novel and possibly crucial associations between perinatal adversity and oxidative stress in both mothers and infants at 6 months. This may indicate a sensitive period during which physiological impact of early environmental risk can be assessed effectively using prospective measures. Future studies will fully evaluate how long-term developmental and oxidative stress status are informed by early maternal infant environment, particularly among mothers experiencing moderate to severe adversity.

## Acknowledgments

We are extremely grateful to all the families for their participation in the studies, as well as their feedback to enhance the study visit experience. We thank research staff members Kathryn Arneaud and Alma Martinez for their efforts on this study.

## Author Contributions

**Conceptualization:** Kameelah Gateau, Lisa Schlueter, Charles A. Nelson, Pat Levitt.

**Data curation:** Lisa Schlueter, Lara J. Pierce, Barbara Thompson, Alma Gharib.

**Formal analysis:** Kameelah Gateau, Ramon A. Durazo-Arvizu.

**Funding acquisition:** Charles A. Nelson, Pat Levitt.

**Methodology:** Kameelah Gateau, Lisa Schlueter, Charles A. Nelson, Pat Levitt.

**Project administration:** Lara J. Pierce, Barbara Thompson, Alma Gharib.

**Supervision:** Charles A. Nelson, Pat Levitt.

**Validation:** Kameelah Gateau, Ramon A. Durazo-Arvizu.

**Writing – original draft:** Kameelah Gateau, Lisa Schlueter.

**Writing – review & editing:** Kameelah Gateau, Lisa Schlueter, Lara J. Pierce, Barbara Thompson, Alma Gharib, Charles A. Nelson, Pat Levitt.

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
