## [Decision Letter · Decision Letter 0]

6 Sep 2023

PGPH-D-23-00615

Exploratory study evaluating the relationships between perinatal adversity, oxidative stress, and infant neurodevelopment across the first year of life

Dear Dr. Gateau,

Thank you for submitting your manuscript to PLOS Global Public Health. After careful consideration, we feel that it has merit but does not fully meet PLOS Global Public Health’s publication criteria as it currently stands. Therefore, we invite you to submit a revised version of the manuscript that addresses the points raised during the review process.

Please be sure to address Reviewer #2's comments on the Introduction and Methods sections, particularly those related to the screening window in the Introduction and all comments in the Methods.

Kindly submit your revised manuscript by 17 October 2023. If you will need more time than this to complete your revisions, please reply to this message or contact the journal office at globalpubhealth@plos.org. Please include the following items when submitting your revised manuscript:

We look forward to receiving your revised manuscript.

Kind regards,

Lisa Miyako Noguchi, PhD

Academic Editor

Journal Requirements:

1. We have noticed that you have a list of Supporting Information legends in your manuscript. However, there are no corresponding files uploaded to the submission. Please upload them as separate files with the item type 'Supporting Information'. 

2. We noticed you have some minor occurrence of overlapping text with the following previous publication, which needs to be addressed:

https://www.researchgate.net/publication/332694118_Childhood_adversity_mental_health_and_oxidative_stress_A_pilot_study

In your revision ensure you cite all your sources (including your own works, and quote or rephrase any duplicated text outside the methods section. Further consideration is dependent on these concerns being addressed.

Additional Editor Comments (if provided):

Reviewers' comments:

Reviewer's Responses to Questions

**Comments to the Author**

1. Does this manuscript meet PLOS Global Public Health’s publication criteria? Is the manuscript technically sound, and do the data support the conclusions? The manuscript must describe methodologically and ethically rigorous research with conclusions that are appropriately drawn based on the data presented.

Reviewer #1: Yes

Reviewer #2: Yes

2. Has the statistical analysis been performed appropriately and rigorously?

Reviewer #1: Yes

Reviewer #2: Yes

3. Have the authors made all data underlying the findings in their manuscript fully available (please refer to the Data Availability Statement at the start of the manuscript PDF file)?

Reviewer #1: Yes

Reviewer #2: Yes

4. Is the manuscript presented in an intelligible fashion and written in standard English?

Reviewer #1: Yes

Reviewer #2: Yes

5. Review Comments to the Author

Reviewer #1: This research is quite relevant to many settings and particularly at these times when newer diagnoses come into existence, but at the same time, it has become relatively difficult to assign/ attribute neurocognitive dysfunction or other diseases, especially those in adults, to a particular cause. Thus, an insight is given to this particular area that can become a point of focus in research and clinical areas.

Generally, issues presented are well articulated, discussions flow are coherent and are in line with objectives and methods proposed.

The quality of figures however, is relatively poor and it is advisable to include clearer images/ figures that will aid the reading.

Reviewer #2: Thank you for the opportunity to review "Exploratory study evaluating the relationships between perinatal adversity, oxidative stress, and infant neurodevelopment across the first year of life." The article focuses on a sample of n=116 mother-infant dyads recruited from primary care practices in Boston and Los Angeles. The article explores the relationship between adversity experienced by women during pregnancy, F2-isoprostane levels in their blood and their infants’ urine, and infant neurocognitive development during various time points in the first year of the infants’ lives. The implications of this research are important, because having a reliable biomarker to predict which infants will benefit from greater neurocognitive supports, and greater family supports, could assist pediatricians and the greater child welfare system to direct resources at higher-risk infants and their families.

Title: the title is appropriate and informative

Introduction: the research questions, claims, and context are clearly explained.

Related research is clearly described and linked to this study.

To make the article accessible to wider audiences, consider clarifying the definition of “internalizing disorders” on page 6, line 30, such as by providing examples of such disorders.

On page 6, lines 146-149, please elaborate on what the existing literature says about the screening window to study the effect of maternal stress on neurocognitive development in offspring, and how it relates to the window chosen for your research study. What evidence is there that the chosen screening window is optimal?

Methods: study design, inclusion and exclusion criteria, and study tools (i.e. EPDS, PHQ9, cumulative risk score) are overall clearly described and sufficiently detailed.

The study design is appropriate for the question being asked.

Mother-infant engagement and family dynamics would have added value to this analysis, and the authors acknowledge this limitation.

On page 9, lines 226-230, consider mentioning what the maximum and minimum score are for the MSEL (minus the gross motor score, as described), for reference.

On page 10, line 235, consider mentioning the factors that can affect F2-isoprostane levels in urine and whether those were accounted for or not, or state that concentration is the only known factor, if appropriate.

Statistics: the statistical analysis is clearly described and appears sound

Results and data: the authors’ results support their conclusions.

Consider elaborating on your interpretation of why higher infant 8-IsoF2a at 2 months is correlated with a lower infant MSEL at 12 moths, but the opposite is true for infant 8-IsoF2a at 6 months and MSEL at 12 months.

Figures and tables: figures and tables are clearly presented.

An editorial note on table 3: there is an inconsistency in the formatting of the r’s and p’s in the table. Some are capitalized and other are not. The tables and figures are otherwise correctly labeled.

References: appear correctly formatted and accurately represented

6. PLOS authors have the option to publish the peer review history of their article (what does this mean?). If published, this will include your full peer review and any attached files.

**Do you want your identity to be public for this peer review?** For information about this choice, including consent withdrawal, please see our Privacy Policy.

Reviewer #1: No

Reviewer #2: **Yes: **Beatrix T Shikani MD MHS FAAP

---

## [Editor Report · Decision Letter 1]

28 Nov 2023

Exploratory study evaluating the relationships between perinatal adversity, oxidative stress, and infant neurodevelopment across the first year of life

PGPH-D-23-00615R1

Dear Dr. Gateau,

We are pleased to inform you that your manuscript 'Exploratory study evaluating the relationships between perinatal adversity, oxidative stress, and infant neurodevelopment across the first year of life' has been provisionally accepted for publication in PLOS Global Public Health.

Best regards,

Lisa Miyako Noguchi

Academic Editor
